# From Feeding Challenges to Oral-Motor Dyspraxia: A Comprehensive Description of 10 New Cases with CTNNB1 Syndrome

**DOI:** 10.3390/genes14101843

**Published:** 2023-09-22

**Authors:** Roberta Onesimo, Elisabetta Sforza, Valentina Trevisan, Chiara Leoni, Valentina Giorgio, Donato Rigante, Eliza Maria Kuczynska, Francesco Proli, Cristiana Agazzi, Domenico Limongelli, Maria Cistina Digilio, Maria Lisa Dentici, Maria Macchiaiolo, Antonio Novelli, Andrea Bartuli, Lorenzo Sinibaldi, Marco Tartaglia, Giuseppe Zampino

**Affiliations:** 1Center for Rare Diseases and Birth Defects, Department of Woman and Child Health and Public Health, Fondazione Policlinico Universitario A. Gemelli IRCCS, 00168 Roma, Italy; roberta.onesimo@policlinicogemelli.it (R.O.); valentina.trevisan@guest.policlinicogemelli.it (V.T.); giuseppe.zampino@unicatt.it (G.Z.); 2Department of Life Sciences and Public Health, Faculty of Medicine and Surgery, Università Cattolica del Sacro Cuore, 00168 Roma, Italy; 3Medical Genetics Unit, IRCCS Bambino Gesù Children Hospital, 00168 Roma, Italy; 4Molecular Genetics and Functional Genomics Unit, IRCCS Bambino Gesù Children’s Hospital, 00146 Roma, Italy; marco.tartaglia@opbg.net

**Keywords:** *CTNNB1* syndrome, NEDSDV, dysphagia, feeding, genotype–phenotype correlation, personalized medicine, transition

## Abstract

*CTNNB1* syndrome is an autosomal-dominant neurodevelopmental disorder featuring developmental delay; intellectual disability; behavioral disturbances; movement disorders; visual defects; and subtle facial features caused by de novo loss-of-function variants in the *CTNNB1* gene. Due to paucity of data, this study intends to describe feeding issues and oral-motor dyspraxia in an unselected cohort of 10 patients with a confirmed molecular diagnosis. Pathogenic variants along with key information regarding oral-motor features were collected. Sialorrhea was quantified using the Drooling Quotient 5. Feeding abilities were screened using the Italian version of the Montreal Children’s Hospital Feeding Scale (I-MCH-FS). Mild-to-severe coordination difficulties in single or in a sequence of movements involving the endo-oral and peri-oral muscles were noticed across the entire cohort. Mild-to-profuse drooling was a commonly complained-about issue by 30% of parents. The mean total I-MCH-FS t-score equivalent was 43.1 ± 7.5. These findings contribute to the understanding of the *CTNNB1* syndrome highlighting the oral motor phenotype, and correlating specific gene variants with clinical characteristics.

## 1. Introduction

Just over a decade ago, de Ligt and colleagues identified loss-of-function variants in the catenin β-1 gene (*CTNNB1*) as the putative genetic event underlying a molecularly unclassified syndromic neurodevelopmental disorder [1]. The neurodevelopmental phenotype characterizing patients carrying heterozygous *CTNNB1* variants was further delineated by Tucci et al. in 2014 [2], who observed that *CTNNB1* syndrome (OMIM #116806) is mainly characterized by developmental delay and intellectual disability (ID) (IQ < 70), and autism spectrum disorder. In addition, affected patients also show muscular hypotonia of the trunk, progressive spastic diplegia, distinctive facial features, together with brain abnormalities, microcephaly (≤2 SD), and mild-to-severe visual impairments [2,3]. In line with these findings, characterization of the ‘batface’ (Bfc) mouse model, resulting from a heterozygous missense variant (p.Thr653Lys) in the C-terminal armadillo repeat of the *CTNNB1* gene, demonstrated a central role of this gene, and β-catenin/cadherins interactions more generally, in neurodevelopment, synaptic plasticity and neuronal network function [2].

Approximately 400 cases, mostly carrying de novo LoF variants, have been described to date [4]. However, due to possible misdiagnosis with cerebral palsy (CP), the actual prevalence is likely underestimated [5].

Given the extremely fragmented knowledge of the condition and paucity of data available, we herein provide a detailed description of 10 patients affected by *CTNNB1* syndrome focusing on their feeding and neurodevelopmental abilities.

## 2. Participants and Procedure

Patients with a confirmed molecular diagnosis of *CTNNB1* syndrome were prospectively recruited at the Rare Disease Unit of the Paediatrics Department, Fondazione Policlinico Agostino Gemelli IRCCS/Medical Genetics Unit, Rome, and Medical Genetics Unit, IRCCS Bambino Gesù Children Hospital, Rome from January 2021 over a period of 24 months. Clinical and demographic information was systematically collected after written informed consent. All study procedures were in line with the Declaration of Helsinki. The Local Ethical Committee (approval n. 2015 of 2019) approved the study as part of a large protocol evaluation on disability and nutritional aspects in rare disease patients.

### 2.1. Clinical History

As a first step, accurate annotation of the identified *CTNNB1* mutation (exon involved, variant, amino acid change, and predicted functional impact) was recorded. Comprehensive personal history was collected in the outpatient setting by pediatricians and clinical geneticists specialized in rare diseases and complex disabilities. Feeding problems, sucking difficulties, and breathing problems during the neonatal period were investigated. A thorough developmental history, including age at which milestones were achieved and current developmental abilities, was collected. Timing related to weaning, the introduction of solid foods and their acceptance were annotated. Detailed questions about patients’ feeding patterns over the course of a typical day and any occurrence of bolus aspiration episodes were asked to primary caregivers, while specific questions on gastroenterological issues were asked by the pediatric gastroenterologist.

### 2.2. Anthropometry

Accurate physical examination was performed in all enrolled subjects. The presence of muscle tone abnormalities was investigated. Occurrence of facial dysmorphism (e.g., broad nasal tip, small alae nasi, long and/or flat philtrum, and thin upper lip vermillion) was systematically assessed by clinical geneticists.

Anthropometric measurements were collected and percentiles calculated through PediTools Electronic Growth Chart Calculators], based on CDC growth charts [6]. Specifically, body mass index (BMI) categorical outcomes were considered ‘overweight’ (BMI ≥ 85th but <95th percentile), ‘obese’ (BMI ≥ 95th percentile), and ‘extreme obese’ (BMI ≥ 120% of the 95th percentile) [7]. Head circumference was measured, and presence of microencephaly was assessed based on CDC growth charts [6].

### 2.3. Observation

A speech-language pathologist directly observed the child’s ability to eat safely and efficiently. Correct execution of the four phases of swallowing was verified. Analysis of airway adequacy and coordination of respiration and swallowing [8] was assessed as well. The observation during mealtime also included the level of independence during feeding, need for supervision or assistance, and ability to use cutlery. Morphodynamics of peri- and endo-oral muscles and structures were examined during simple to more complex movements. Secretion management and presence of childhood apraxia of speech was monitored [9].

### 2.4. Scale and Questionnaire

Secretion management was quantified through the Drooling Quotient 5 (DQ5) [10,11]. Specifically, DQ5 represents a semiquantitative, direct observational method that evaluates drooling by measuring leaked saliva from the mouth. During 5 min, for every interval of 15 s, the presence or absence of drooling was determined [10].

The Italian version of the Montreal Children’s Hospital Feeding Scale (I-MCH-FS) [12] was administered to assess the presence of feeding disorders. The scale is composed of 14 items covering five domains related to mealtimes: parental concerns about feeding and children’s growth; strategies; children’s oral motor and sensory abilities; appetite, mealtime duration, and behavior; and family relationships influenced by feeding [13].

## 3. Results

Ten unrelated participants (8 M; mean age 10.8 ± 6.3 y; median age 9 y; and age range 6–23 y) were included in this study. All subjects showed *CTNNB1* haploinsufficiency due to pathogenic/likely pathogenic variants (six nonsense and three frameshift) with the exception of one deletion, involving multiple exons (case 4). The well-recognized c.1759 C>T p.(Arg587Ter) variant [14] was the most frequent variant among our cohort identified in three individuals carriers (Table 1).

### 3.1. Clinical Features

#### 3.1.1. Feeding Abilities

##### Prenatal and Neonatal Period

For the whole cohort, during pregnancy, amniotic fluid disorders were not reported and no premature births occurred. Delivery was uneventful in six cases, while four newborns presented perinatal asphyxia/respiratory distress. Two of them required nutritional support by the placement of a transient nasogastric tube (NG-tube). In addition to the two tube-fed cases, one female newborn showed incompetence in breastfeeding with fatigability, managed through the use of a special bottle. Overall, at birth, nutritive sucking skills were adequate for seven newborns out of the whole cohort (70%). During the neonatal period, floppiness was a constant finding in our cohort (100%) (Table 2).

##### Infancy

Weaning was regularly performed in most cases (90%) at a mean age of 5.4 months, except for one case in whom weaning started at 9 months of age. Instead, the proposal of the first chewable soft solid was timely performed at 10–12 months of life in only 60% of cases. For the whole cohort, the mean age for the achievement of this milestone was 18 months. Specifically, soft solid foods were introduced into the diet at 24 months in 20% of cases (*n* = 2/10) and at 36 months in a further 20% of cases (*n* = 2/10). The late introduction of solid foods was associated with a preference to smooth pureed food textures and constant episodes of refusal to eat lumpier foods (Table 2).

##### Childhood

In one subject, frequent episodes of liquids aspiration together with choking episodes with solids foods were reported in the first two years of life and underwent progressive resolution over time.

At the time of the current evaluation, 100% of participants were able to safely swallow semisolid and small pieces of solid foods, and 90% of them were able to safely swallow liquids. Specifically, a 4.5-year-old child showed poor coordination in swallowing liquids, especially when using a bottle or straw rather than a glass.

With respect to chewing ability, all parents reported mastication to be mostly fast and vertical (100%). In terms of food taste selectivity, three children were reported to have a stronger tendency to eat a limited range of flavors (30%). Two of them demonstrated persistent over-selectivity in their eating patterns during their whole childhood, without any improvement over time (see Table 2).

#### 3.1.2. Gastrointestinal Issues

The clinical interview revealed that during the first six months of life only one child suffered from gastro-esophageal reflux (GERD). During childhood, one child suffered from GERD for 3 years (see Table 2).

In seven participants out of the whole cohort (70%), the mean frequency of bowel movements was 1–2 times every 1–2 days. For the remaining three patients (30%), evacuations were less frequent, sometimes painful, accompanied by abdominal pain, frequently requiring the use of enemas or laxatives.

#### 3.1.3. Development Milestones

During infancy, the evidence of developmental milestone delay along with distinctive facial features raised the need for genetic counselling. The ability to hold the head steady without support was achieved at a mean age of 8.5 months (range 3–24 months). Trunk control was reached at a mean age of 18 months (range 9–48 months). Instead of crawling, 9 out of 10 children tended to slide from one place to another.

With growth, hypotonia of the trunk and hypertonia of the legs manifested in the whole cohort. Seven children were able to walk independently, even just for a few meters, at a mean age of 3.5 years (range 2–8 years), while the remaining three required the support of a walker. One child faced a regression in walking ability when she was 6 years old due to the interruption of psychomotricity sessions and physical rehabilitation treatments during the COVID-19 pandemic.

Language development was similarly delayed. The first words were spoken on average at around the age of 3 years (age range: 1–4 years). At the time of the current evaluation, half of the patients (*n* = 5/10) were not able to build sentences and speak fluently. The remaining 50% started to produce sentences on average upon reaching the age of 5 years (age range: 3–8 years). The majority of patients (*n* = 7/10) were entered early into a speech-language rehabilitation program aimed at improving communicative language skills; however, one patient reported no significant improvement. One male child was diagnosed with autism spectrum disorder (ASD). A variable level of ID (from mild-to-severe) was universally detected.

#### 3.1.4. Anthropometry

Only in one case, intrauterine growth retardation had been diagnosed during the eighth month of gestation, resulting in a low weight at birth. For the remaining nine newborns (90%), birth weight fell within the normative ranges.

The craniofacial phenotype comprised microcephaly (ranging from 2 to 5 SD) in 7 out of 10 patients. Weight and height measurements revealed a BMI in the normality range for six cases. Two participants were overweight, and two were underweight (Appendix A).

### 3.2. Observation

Mild-to-severe coordination difficulties in single or in a sequence of movements involving the endo-oral and peri-oral muscles were noticed across the entire cohort. Following the request for specific oral-motor tasks (e.g., whistling or clucking the tongue), the performance lacked preciseness and accuracy. Spatial (e.g., larger movements) and temporal (e.g., delated initiation) errors were noted. The observation of oral-facial praxis revealed difficulties in oral-facial planning skills, more commonly for complex movements rather than for simple single movements.

Through meal observation with solid foods by an experienced speech language therapist, the jaw pattern was found to be poorly controlled. In no patient, a mature chewing pattern in terms of precise coordination and strength of movements was observed. A circular rotary jaw movement was substituted with a limited lateral movement, often accompanied by tongue thrusting. Movements of the cheeks, lips, and tongue were coarsely controlled. A weak oral competence with a consequential liquid loss was also observed. Fine tongue movements during the preparatory phase of swallowing of the tongue were lacking. In one subject, unintentional leakage of the liquid bolus into the pharynx (with subsequent coughing) was indirectly suspected after poor control by oral muscles. The use of age-appropriate handling of cutlery and glass was not achieved by the majority of patients.

Phonetic disorders were universally observed during an unstructured conversation with patients, with mild-to-severe articulatory deficits. Consecutive articulatory movements requiring rapid execution were impaired at speed. Fluency and intelligibility of speech was only sometimes achieved. Stereotyped emissions of guttural sounds were observed in a 6-year-old child.

### 3.3. Scale and Questionnaire

Mild-to-profuse drooling was a commonly complained-about issue by 30% of parents (*n* = 3/10), with DQ5 values of more than 18.

The I-MCH-FS was administered to the whole cohort. The mean total t-score equivalent was 43.1 ± 7.5 (range 36 to 55). On average, mealtimes with the child were considered to be easy (item #1) and not involve any concerns in terms of timing or bad behaviors during mealtimes (item #6). Distractors including toys or smartphones were often needed to encourage eating (item #9). Chewing ability was considered improvable (item #11). In one case, the need of continuous supervision, along with a great concern for the child eating (item #2), was reported due to previous choking episodes. Most responders considered their child’s growth to be adequate (item #12), although in three cases, they reported better growth in length than in weight (see Table 3).

## 4. Discussion

The development of next generation sequencing techniques (NGS), such as whole-exome sequencing (WES), has improved the ability to identify causative gene variants even for patients with unexplained neurodevelopmental disorders lacking a striking clinical phenotype [15,16,17]. The *CTNNB1* syndrome is a recently described autosomal dominant neurodevelopmental disorder caused by de novo LoF variants in the *CTNNB1* gene [18]. It is usually suspected in individuals with suggestive findings, such as developmental delay, intellectual disability, autistic behaviors, and spastic diplegia [5]. Patients may encounter feeding challenges from birth and throughout their entire lifespan. Although we have highlighted how eating issues do not compromise everyday life in a severe manner, complex feeding skills remain challenging skills to acquire for such patients.

In contrast to what has been observed in other pediatric genetic conditions [19], parents of children with *CTNNB1* syndrome comprehensively had a subjective impression of harmonious growth, as detected by I-MCH-FS administration and confirmed by anthropometric findings, in accordance with previous data [20]. Additionally, although the clinical features of *CTNNB1* syndrome overlap with CP, *CTNNB1*-mutated patients have a reduced risk of oropharyngeal dysphagia [21], which might be relevant in the differential diagnosis between the two conditions.

From this standpoint, we can illustrate how the rate of sucking difficulty at birth is limited to 30% of patients. Few other cases of feeding difficulties at birth have been previously described. For instance, Verhoeven et al. reported marked feeding difficulties at birth in a patient carrying a de novo splice site variant in the *CTNNB1* gene (c.734+1G>T) associated with microcephaly, short stature, low weight, and developmental delay [22]. Hypotonia and poor sucking with the exclusion of breastfeeding was observed in a Bangladesh female newborn carrying a de novo *CTNNB1* nonsense mutation (c.1420C>T, p.Arg474Ter). The same child in her childhood developed oral stereotypic movements including tongue protrusion, scarce hungriness, and immature chewing [23]. The emergence of marked feeding difficulties and hypotonia was reported also for an Italian–Philippine male newborn with a de novo nonsense variant (p.Cys419Ter) in exon 9 of the *CTNNB1* gene, suffering from neonatal seizures and presenting brain anomalies, including corpus callosum dysgenesis and hypoplastic brain stem [24].

In our present cohort, poor sucking–swallowing–breathing coordination was managed and overcome by a dedicated team of experts with the intention of transitioning from enteral feeding to safe oral feeding with an adequate caloric intake. In accordance with previous studies reporting only one individual carrying a de novo splice mutation c.1081+1G>C in the intron 7 of the *CTNNB1* gene and requiring prolonged enteral feeding due to severe failure to thrive [23], none of our patients required a G-tube placement.

Furthermore, we found that while weaning in the first year of life smoothly occurred in most cases (90%), the introduction of solid foods caused concern. In our cohort, only 60% of children timely accepted the introduction of solid foods into their diet. Evidence shows how during early childhood, food fussiness and picky eating is typical and new food has to be proposed up to 15 times for acceptance in the general pediatric population [24,25,26]. Nevertheless, persistence of refusal is to be considered a red flag to investigate. Of relevant interest is how the failure to timely introduce more sophisticated food tastes and textures negatively impacts the development of proper chewing skills [27].

Interestingly, in contrast to other reported conditions in which a slow but gradual improvement in the acquisition of complex feeding skills is appreciable [28], in the present cohort of patients, a mature chewing pattern was never developed. In addition, we noticed a scarce coordination in fine movements of the oral-facial muscles not only during the preparatory phase of swallowing, but also during the execution of oral-facial praxis and speech. These findings suggest that a more specific praxis deficit might exist. Although in *CTNNB1* syndrome oral-facial dyspraxia has been anecdotally reported [20,23,29,30], mastication dyspraxia has not been described to date. This term was first used by Mariën et al. (2013) [31] in reporting the chewing pattern of a 5-year-old patient diagnosed with a developmental coordination disorder. The attempts to chew were described as ‘rough, effortful, and laborious biting movements confined to the vertical plane’ [31] not having a ‘teardrop shape’ [32]. Specifically, the ‘teardrop shape’ involves a jaw displacement towards the mastication side during the opening phase and the opposite side during the closing phase with a slight rotatory movement. As the bolus becomes smaller, the vertical width of the movements progressively decreases [32]. In accordance with Mariën et al. (2013)’s description [31], chewing patterns observed in the present cohort were similar and lacked fine motor coordination of the teeth, jaw, and facial muscles, occasionally leading to aspiration episodes.

Due to the lack of curative treatments for *CTNNB1* syndrome, supportive care from birth includes, among others, a physical therapist, occupational therapist, speech-language pathologist, gastroenterologist, and expert pediatricians and counsellors. Although clinical practice guidelines have yet to be published for this condition [5], prompt management of feeding difficulties is suggested [33,34]. Feeding management in *CTNNB1* syndrome, similar to other complex conditions [19,35], begins from birth and accompanies the patient throughout growth. In this context, treatments supporting children with *CTNNB1* syndrome in achieving the milestones are fundamental to establish the prerequisites for safe swallowing, including head control and correct head–trunk–pelvis alignment [36], often compromised by abnormal muscle tone. Later on, rehabilitation has to be guaranteed in order to develop and master challenging feeding tasks, including the independent use of cutlery often compromised by peripheral hypertonia/spasticity.

Due to the risk of dyspraxia, which may result in aspiration episodes, targeted treatments should be guaranteed. This latter finding, reported in our cohort as well as in previous studies [23], negatively impacts patients’ quality of life [37]. Treatment options for developmental dyspraxia have been discussed in the medical literature [38]. Proper rehabilitation outcomes, however, may be compromised by behavioral disorders, including ASD or aggressive/destructive behaviors [5]. Moreover, in one case, we highlight how discontinuation of treatment for prolonged periods can negatively impact on maintenance of achieved abilities. A clear deterioration during the COVID-19 pandemic owing to the inability to continue usual treatments has been reported in both pediatric and adult populations [39,40].

Considering the oro-motor compromission, case 4 carrying the multi-exon deletion showed the most severe phenotype with dysphagia for solids and liquids. A moderate phenotype characterized only by liquid dysphagia was observed in case 5, carrying the variant in exon 9. Considering oro-motor dyspraxia, our data expand and further characterize the genotype–phenotype association proposed by Mirosevic et al. [18], suggesting that variants in exon 9 (case 5) and 11 (case 2,3 and 10) might be associated from a mild-to-moderate phenotype. Although, given the limitations of our study (i.e., the small cohort size and a detailed focus on oral-motor-dyspraxia), more evidence is needed to unravel a clear genotype–phenotype correlation.

## 5. Conclusions

In summary, our study contributes to the understanding of the *CTNNB1* syndrome highlighting the oral-motor phenotype and correlating specific genetic variants with clinical characteristics. Overall, our observation suggests that *CTNNB1*-mutated patients might show a reduced risk of oropharyngeal dysphagia and oral-motor dyspraxia, but struggle in developing a mature chewing pattern. On the other hand, patients carrying a multi-exon deletion are associated with a more severe phenotype in terms of feeding and swallowing abilities.

These findings enhance the diagnostic and clinical management approaches for individuals with *CTNNB1* syndrome, emphasize the importance of early recognition via genetic testing, and suggest the need for multidisciplinary care to optimize outcomes and personalized treatments.

## 6. Limits and Future Research

The limited number of patients included in our cohort requires that specific further observations are needed. Moreover, the age range of our cohort is limited to infancy, which also provides an opportunity for future research to better characterize the clinical history of the syndrome during adulthood. Further research and collaborations are warranted to advance our knowledge on genotype–phenotype correlations of this rare syndrome and develop targeted interventions.

## Figures and Tables

**Table 1 genes-14-01843-t001:** Main clinical features and genetic findings in our cohort of 10 *CTNNB1* patients.

PT ID	1	2	3	4	5	6	7	8	9	10
**Age (y)**	6	6	23	15.5	4.5	5.5	10.5	10	9	10.5
**Eye abnormalities**	+	+	+	+	+	+		+	+	+
**Muscle tone** **abnormalities**	+	+	+	+	+	+	+	+	+	+
**Microcephaly**	+			+	+		+	+	+	+
**MRI anomalies**	+	+							+	
**Autism spectrum disorder**							+			
**Speech difficulties**	+	+	+	+	+	+	+	+	+	+
**Intellectual ** **Disability/** **developmental ** **delay**	+	+	+	+	+	+	+	+	+	+
**Motor milestone delay**	+	+	+	+	+	+	+	+	+	+
**Abnormal BMI**		+		+	+				+	
**Low weight at birth**	+									
**Perinatal asphyxia/** **respiratory distress**				+	+			+	+	
**CTNNB1 [NM_001904.3] variants **	c.680dup, p.Leu299ThrfsTer5	c.1759C>T, p.Arg587Ter	c.1759C>T, p.Arg587Ter	c.1081+1_1082-1_2346+?del	c.1420C>T, p.Arg474Ter	c.998dupA, p.Tyr333Ter	c.975delA, p.Asn326IlefsTer2	c.1874del, p.Lys625fs	c.976_979delAATA, p.Asn326Ter	c.1759C>T, p.Arg587Ter
**Variant type**	frameshift	nonsense	nonsense	deletion	nonsense	nonsense	frameshift	frameshift	nonsense	nonsense
**Exon involved**	5	11	11	exons 9 to 16 (deletion)	9	7	7	12	7	11

BMI = body mass index; MRI = Magnetic Resonance Imaging; Pt = patient; y = years. “+” can be explained as “presence” or “occurence”.

**Table 2 genes-14-01843-t002:** Main feeding and gastrointestinal issues in our cohort of 10 cases with *CTNNB1* syndrome.

Pt ID	1	2	3	4	5	6	7	8	9	10
**Current age (y)**	6	6	23	15.5	4.5	5.5	10.5	10	9	10.5
**Birth**										
** *Asphyxia/respiratory distress* **				+	+			+	+	
**Peri and neonatal period**										
** *Poor sucking* **				+			+			+
** *Tube feeding* **				+			+			
** *GER* **					+					
**Infancy**										
** *Delayed weaning* **									+	
** *Delayed introduction of solid foods* **				+				+	+	+
**Childhood**										
** *Food texture selectivity* **				+				+	+	+
** *Food taste selectivity* **				+					+	+
** *Dysphagia for solid* **				+						
** *Dysphagia for liquids* **				+	+					
** *Immature chewing* **	+	+	+	+	+	+	+	+	+	+
** *Drooling* **	+	+	+			+				
** *GER* **								+		
** *Constipation* **	+				+				+	

**Table 3 genes-14-01843-t003:** The I-MCH-FS equivalent scores in our cohort of *CTNNB1*-syndrome patients.

Individual Item	Cohort (N = 10)	
	Mean ± SD	Median
*1 How do you find mealtimes with your child?*	1.8. ± 1.7	1.0
*2 How worried are you about your child’s eating?*	1.3. ± 0.9	1.0
*3 How much appetite (hunger) does your child have?*	1.3 ± 0.9	1.0
*4 When does your child start refusing to eat during mealtimes?*	1.6 ± 1.8	1.0
*5 How long do mealtimes take for your child (in minutes)?*	2.3 ± 0.6	2.0
*6 How does your child behave during mealtimes?*	1.9 ± 1.9	1.0
*7 Does your child gag or spit or vomit with certain types of food?*	1.9 ± 1.9	1.0
*8 Does your child hold food in his/her mouth without swallowing it?*	1.6 ± 1.8	1.0
*9 Do you have to follow your child around or use distractions (toys, tv) so that your child will eat?*	3.4 ± 2.9	1.0
*10 Do you have to force your child to eat or drink?*	1.3 ± 0.9	1.0
*11 How are your child’s chewing (or sucking) abilities?*	2.1 ± 1.3	1.5
*12 How do you find your child’s growth?*	1.3 ± 0.9	1.0
*13 How does your child’s feeding influence your relationship with him/her?*	1.0 ± 0.0	1.0
*14 How does your child’s feeding influence your family relationships?*	1.0 ± 0.0	1.0
*Total*	43.1 ± 7.5	39.5

## Data Availability

The datasets generated during the current study are available from the corresponding author on reasonable request.

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
