# Peer review of "From Feeding Challenges to Oral-Motor Dyspraxia: A Comprehensive Description of 10 New Cases with CTNNB1 Syndrome"

_genes, 2023, doi:10.3390/genes14101843_

Round 1

Reviewer 1 Report

Dear authors. Congratulations on the initiative to carry out this work. Unfortunately, the article suffers from several important weaknesses that need to be addressed before the paper can be considered for publication:

1. The abstract is unclear, has formatting problems and has several XXXXXX.

2. Where the name of the Bioethics Committee should be, there is a XXXXXXXXXXXXXXXX.

3. In general, the wording of the article is messy and not easy to read.

4. The tables and graphs are poorly formatted and out of order. 

5. A major problem with the article is that no genotype-phenotype correlation is presented. With the patients presented it is possible and necessary to do a genotype-phenotype correlation. 

I hope these criticisms are constructive and serve to improve your work.

Author Response

Dear Reviewer,

Thank you for your helpful reviews.

In the updated version of the paper, we revised the formatting issues and revised the abstract.

Furthermore, the name of the Ethical Committee and the protocol number have been specified.

A proof-read of the article was performed by an English native speaker and the manuscript wording was ameliorated.

Tables and graphs were better formatted, in order to be more clear and easy to read.

Finally, in the updated manuscript we included some fundamental considerations involving genotype-phenotype correlations. In particular, we observed a mild to moderate phenotype in terms of oro-motor dyspraxia in patients carrying variants in exon 9 (case 5) and 11 (cases 2-3 and 10). However, additional studies involving larger cohorts of patients are needed to confirm our findings.

King regards

Reviewer 2 Report

Dear authors,

thank you for this interesting manuscript on CNNTB1 related disorders. You present novel patients in a new context, albeit the disorder had previously been reported extensively (but not in this perspective!). There indeed is novelty in your report. 
There a several ethical concerns as to where and with which protocol you recruited the patients (currently noted as xxx). This by itself warrants a major revision as it is a significant point without which I would not recommend publishing. 
In general, the methods are sound and the incremental novelty of this manuscript is important to the further understanding of this disease as well as the following disease management as well as genetic counselling. 

There are only few errors in English language, so I propose to have it proofread by a native speaker. 

Author Response

Dear Reviewer,

Thank you for precious comments.

We revised the paper accordingly, specifying the protocol number and the Institutions involved in the current study.

Additionally, a proof-read of the article was performed by an English native speaker and the manuscript wording was ameliorated.

King regards

Round 2

Reviewer 1 Report

Congratulations to the authors for their excellent work and for making the changes requested in the first revision clear and well made. This study can be published in its present form. 

Author Response

The thank the Reviewer for her/his positive comments. 

Reviewer 2 Report

Dear authors,

again, I applaud you for addressing an important topic. I merely have some minor comments.

1. "All subjects showed CTNNB1 haploinsufficency due to de novo loss of function (LoF) variants (6 nonsense and 3 frameshift) with the exception of one deletion, involving multiple exons (case 4). " (ll. 124-125)

As there are no Western Blots or qPCR in this study, it would be speculative to denote the varints as either haploinsufficiency or loss of function. I would delete all mentions of that and instead just denote the variants as likely pathogenic.

2. The table denotes the genetic variants in the row "Ctnnb1 [nm_001904.3] variants". Please change to "CTNNB1 [NM_001904.3]".

3. The table denotes the variant in patient 4 as "c.1081+1_1082-1_2346+?del" but the text rather mentions "c.1081+1G>A". Could you please specify?

4. "tended to move slighterly from one place to another." (l.180)

Did you perhaps mean "slightly"?

5. "From this standpoint, we can evidence how the rate of sucking difficulty" (l.260)

I would recommend to change to "...can illustrate how..."

6. "target treatments" (ll.317-318)

please change to 'targeted treatments'

7. "patient's quality of life" (l.319)

please change to "patients' quality of life"

Best wishes

Author Response

Reviewer 2’s Comments:

Dear authors,

again, I applaud you for addressing an important topic. I merely have some minor comments.

 AA: We thank the reviewer for her/his positive comments. All the suggested correction have been implemented.

1."All subjects showed CTNNB1 haploinsufficency due to de novo loss of function (LoF) variants (6 nonsense and 3 frameshift) with the exception of one deletion, involving multiple exons (case 4). " (ll. 124-125). As there are no Western Blots or qPCR in this study, it would be speculative to denote the varints as either haploinsufficiency or loss of function. I would delete all mentions of that and instead just denote the variants as likely pathogenic.

AA: Modified as requested.

  1. The table denotes the genetic variants in the row "Ctnnb1 [nm_001904.3] variants". Please change to "CTNNB1 [NM_001904.3]".

AA: Modified as requested.

  1. The table denotes the variant in patient 4 as "c.1081+1_1082-1_2346+?del" but the text rather mentions "c.1081+1G>A". Could you please specify?

AA: We checked the table and text for in correctness.

  1. "tended to move slighterly from one place to another." (l.180)

Did you perhaps mean "slightly"?

AA: We modified the sentence for clarity.

  1. "From this standpoint, we can evidence how the rate of sucking difficulty" (l.260)

I would recommend to change to "...can illustrate how..."

 AA: We changed the sentence for clarity.

  1. "target treatments" (ll.317-318)

please change to 'targeted treatments'

  AA: We changed the sentence for clarity.

  1. "patient's quality of life" (l.319)

please change to "patients' quality of life"

  AA: We changed the sentence for clarity.

Best wishes

So, in the end, we sincerely hope that all corrections made might be considered valid for accepting our paper by Genes.

We will be waiting for your final decision and we sincerely give you our personal best regards,

I confirm that we choose the "Open Review" option.

The corresponding author & co-authors